# Standard Doses of Cholecalciferol Reduce Glucose and Increase Glutamine in Obesity-Related Hypertension: Results of a Randomized Trial

**DOI:** 10.3390/ijms25063416

**Published:** 2024-03-18

**Authors:** Catarina Santos, Rui Carvalho, Ana Mafalda Fonseca, Miguel Castelo Branco, Marco Alves, Ivana Jarak

**Affiliations:** 1Health Sciences Faculty, University of Beira Interior, 6200-505 Covilhã, Portugal; mcbranco@fcsaude.ubi.pt; 2Nephrology Department, Castelo Branco Local Health Unit, 6000-128 Castelo Branco, Portugal; 3Department of Life Sciences, Faculty of Sciences and Technology, University of Coimbra, 3000-456 Coimbra, Portugal; carvalho@bioq.uc.pt; 4CICS-UBI, Investigation Center for Health Sciences, University of Beira Interior, 6200-505 Covilhã, Portugal; mfonseca@fcsaude.ubi.pt; 5Laboratory of Endocrine and Metabolic Research, UMIB—Unit for Multidisciplinary Research in Biomedicine, ICBAS—School of Medicine and Biomedical Sciences, University of Porto, 4050-313 Porto, Portugal; alvesmarc@gmail.com; 6Laboratory for Integrative and Translational Research in Population Health (ITR), University of Porto, 4200-465 Porto, Portugal; 7Laboratory of Physiology, Department of Immuno-Physiology and Pharmacology, ICBAS—School of Medicine and Biomedical Sciences, University of Porto, 4050-313 Porto, Portugal; 8Laboratory of Drug Development and Technologies, Faculty of Pharmacy, University of Coimbra, 3000-548 Coimbra, Portugal; jarak.ivana@gmail.com; 9i3S-Institute for Investigation and Health Innovation, University of Porto, 4200-393 Porto, Portugal

**Keywords:** vitamin D, hypertension, glutamine, obesity, metabolites, glucose

## Abstract

In arterial hypertension, the dysregulation of several metabolic pathways is closely associated with chronic immune imbalance and inflammation progression. With time, these disturbances lead to the development of progressive disease and end-organ involvement. However, the influence of cholecalciferol on metabolic pathways as a possible mechanism of its immunomodulatory activity in obesity-related hypertension is not known. In a phase 2, randomized, single-center, 24-week trial, we evaluated, as a secondary outcome, the serum metabolome of 36 age- and gender-matched adults with obesity-related hypertension and vitamin D deficiency, before and after supplementation with cholecalciferol therapy along with routine medication. The defined endpoint was the assessment of circulating metabolites using a nuclear magnetic resonance-based metabolomics approach. Univariate and multivariate analyses were used to evaluate the systemic metabolic alterations caused by cholecalciferol. In comparison with normotensive controls, hypertensive patients presented overall decreased expression of several amino acids (*p* < 0.05), including amino acids with ketogenic and glucogenic properties as well as aromatic amino acids. Following cholecalciferol supplementation, increases were observed in glutamine (*p* < 0.001) and histidine levels (*p* < 0.05), with several other amino acids remaining unaffected. Glucose (*p* < 0.05) and acetate (*p* < 0.05) decreased after 24 weeks in the group taking the supplement, and changes in the saturation of fatty acids (*p* < 0.05) were also observed, suggesting a role of liposoluble vitamin D in lipid metabolism. Long-term cholecalciferol supplementation in chronically obese and overweight hypertensives induced changes in the blood serum metabolome, which reflected systemic metabolism and may have fostered a new microenvironment for cell proliferation and biology. Of note, the increased availability of glutamine may be relevant for the proliferation of different T-cell subsets.

## 1. Introduction

Hypertension is a multifactorial disease resulting from the dynamic interaction of an array of factors, including, but not limited to, genetic, hemodynamic, dietetic, environmental, and lifestyle factors. It is frequently associated with several other chronic conditions, such as obesity and metabolic syndrome, and is characterized by a series of metabolic changes in the various biological compartments, as well as oxidative stress and inflammation. While a number of drugs are used to treat hypertension, all of them have their shortcomings; it is especially difficult to treat resistant and refractory patients. The recently discovered involvement of the immune system [1] has raised the possibility that other drugs, such as immunomodulators, could be effective at treating hypertension. However, most of the immunosuppressants currently used are not acceptable for treating hypertension from the clinical risk point of view. This is in contrast with other conditions, such as transplantation, in which the benefit clearly outweighs the risks. To circumvent this, safer alternatives are emerging as potential therapeutic options [2,3]. One such option may be vitamin D. Its role as an effective modulator of the immune system has gained much attention in recent years, especially in the prevention and treatment of various diseases [2,3].

In the case of cholecalciferol (i.e., vitamin D3), besides its immunomodulatory properties [4,5], the ability to inhibit the mechanistic target of rapamycin (mTOR) activity is also very relevant. mTOR regulates many cellular processes related to proliferation and metabolism, and changes in the availability of several micronutrients, including amino acids and/or glucose [6,7], may affect mTOR activity and cell reprogramming. In this way, vitamin D-induced serum metabolite modulation may be of utmost importance in the mitigation of the effects of chronic inflammation or end-organ disease [8]. In this setting, a substantial part of the evidence concerns glucose metabolism and glycolysis inhibition [9], but vitamin D can also promote anabolic pathways [10] via mTOR downregulation [10].

However, the translation of in vitro results into clinical outcomes is not straightforward. Across trials of vitamin D supplementation, the results are not homogeneous and are sometimes dependent on the patient population studied [11,12]. Specifically, the effects of supplementation at the level of the serum metabolome in obese hypertensives are not known. Evidence from other clinical conditions points to reduced oxidative stress in multiple sclerosis [13], no effect in prediabetes [14], and a high correlation with lipid metabolism [15,16]. Some trials also document that vitamin D may represent an important anti-inflammatory mediator in the setting of obesity [17] and that even short periods of treatment may induce significant changes in the serum metabolomes of acutely ill patients [18], cystic fibrosis patients [19], or peripheral immune cells [20,21]. On the other hand, insufficient levels of vitamin D are frequently observed in hypertension and are further aggravated in the setting of obesity, leading us to hypothesize that vitamin D can have a role as a disease modifier and a player in metabolism. The most frequently described metabolic pathways modulated by vitamin D include glycolysis/gluconeogenesis, lipid metabolism (including fatty acids and glycosphingolipids), and urea cycle metabolism [19].

Here, we present an expanded dataset from a previously described cohort in which we analyzed the immunomodulatory effects of vitamin D [22]. Since immune cells depend on the nutrients available in their environment, it is plausible that modifications in the serum metabolome can affect immune cell populations.

In this regard, we aimed to explore the effects of cholecalciferol on the serum metabolome of hypertensive patients to evaluate which pathways are most affected by therapeutic intervention and if an anabolic effect can be observed. To that end, normo- and hypertensive patients with underlying obesity or metabolic syndrome were analyzed after cholecalciferol supplementation. Moreover, given that high-fat diets in combination with other diseases could potentiate the dysregulation of numerous pathways of amino acid and lipid metabolism [23], we hypothesized that obese hypertensives could represent a good model to track the effects of interventions aimed to modulate the serum metabolome.

## 2. Results

### 2.1. Study Participants and Normotensive Controls

The trial recruited thirty-six hypertensives. Seven had metabolic syndrome, and the remaining participants were obese. Eighteen patients were randomized for each group and allocated to the intended treatment (CONSORT flow chart diagram (Appendix A). All participants were Caucasian. The average age, body mass index (BMI), and waist circumference for all participants were 57.1 ± 5.5 years, 34.5 ± 5.1 Kg/m^2^, and 113.8 ± 10.7 cm, respectively. Data regarding average values for each of the baseline groups are presented in Table 1.

The recruited participants were prescribed all available classes of anti-hypertensives, but inhibitors of the renin–angiotensin system were the most frequently documented drugs. Seven patients had four anti-hypertensives prescribed. At baseline, the average systolic and diastolic blood pressures for the entire cohort were 135.6 ± 18.2 mmHg and 81.9 ± 10.2 mmHg, respectively. Severe adipose tissue dysfunction, as measured using the visceral adiposity index, was high for all participants, with an average value of 5.7 ± 3.8. During the trial, one patient was non-compliant with the trial medication, but no losses to follow-up were documented.

The normotensive controls (N = 17), seven of which were women, had an average BMI of 27.8 ± 4.1 Kg/m^2^. The age range for this group was the same as for the trial participants. None of the normotensive controls had a history of diabetes, and measurements of glycated hemoglobin (HbA1c) were not performed for this group. Only a baseline sampling for blood chemistry and metabolomic analysis was obtained with no additional interventions (Table 2).

### 2.2. Variation in Vitamin D and Free Vitamin D

For the control group, no changes were observed between weeks 0 and 24, but in the active treatment group, supplementation resulted in a significant increase in vitamin D (26.9 ± 5.6 ng/mL) (Table 3).

By week 16, a peak in the average vitamin D concentration was documented for the cholecalciferol group, with an average value of 39.4 ± 7.5 ng/mL being registered. According to the protocol, the transition to a maintenance dose was carried out, but this resulted in a drop in the vitamin D levels. By the end of the trial, the average value of vitamin D for treated patients was under 30 ng/mL. Even if the baseline vitamin D levels almost doubled, the range was still at a suboptimal level. For free vitamin D, an increase was also documented in the treated patients, from 5.2 ± 1.2 pg/mL to 6.2 ± 2.3 pg/mL (*p* < 0.001). Since precise cutoff levels for serum free vitamin D sufficiency are not yet available, we followed the manufacturer’s instructions. Levels above 5.8 pg/mL are sufficient, as a rule, and are equivalent to total vitamin D ≥ 30 ng/mL. Thus, we may consider the levels obtained in the active treatment group sufficient at the end of the trial, at least for free vitamin D, the fraction that is biologically active and available. However, no changes in blood pressure were documented in the trial participants between the two time points with increasing values of vitamin D (Table 3).

### 2.3. Serum Metabolome of Normotensive Controls and Hypertensive Trial Participants

Clustering tendencies and possible outliers in metabolome profiles collected at different time points were analyzed using principal component analysis (PCA). Since for the hypertensive trial participants no differences in metabolic profiles were found at time 0, they were treated as one group (Hyper0). Comparison of the metabolic signatures of the normotensive controls (Normo0) and hypertensive patients (Hyper0) collected before vitamin D therapy revealed clustering tendencies according to the tested groups. Although a certain overlap could be observed in the PCA model, group separation between the Normo0 controls and Hyper0 patients indicated metabolome differences (Figure 1A).

Further statistical models were applied to investigate these differences. An orthogonal projection to latent structures discriminant analysis (OPLS-DA) model used to increase group separation and identify differentially expressed metabolites demonstrated that the two groups were separated without overlap (Figure 1B), with high goodness of fit and predictability confirmed by the model validation based on random permutation (Appendix A).

Variable importance-to-projection values (VIPs) were extracted for each variable, and those with a VIP > 1 and a univariate *p* < 0.05 were considered to be overexpressed metabolites that contributed to the difference between the groups (Figure 2).

Differences in various classes of metabolites were characteristic of hypertension, including amino acids and those related to glucose and lipid metabolism (lactate and acetate). Overall, decreased expression of several amino acids, including glycine, leucine, glutamine, lysine, alanine, and the aromatic amino acids phenylalanine and tyrosine, was documented in hypertensives. Additional metabolites were only identified as differentially expressed by the univariate analysis, including, amongst others, glucose (Appendix A), an expected finding given the prevalence of diabetic patients in the cohort of the trial participants. Nevertheless, the average glycated hemoglobin levels were within the recommended target of <7%, and the highest value observed amongst all participants was 6.6%.

### 2.4. Serum Metabolome after Cholecalciferol Supplementation

To investigate the possible effects of vitamin D therapy, we performed pairwise comparisons of hypertensive participants before (Hyper-AG0) and after (Hyper-AG24) cholecalciferol supplementation. Thirty-one metabolites were analyzed and included in the PCA model. Due to spectral artifacts and compromised spectral quality, 16 paired values were included in the analysis (with the two remaining pairs being excluded). As predicted in Figure 3A, a high degree of overlap existed between weeks 0 and 24. Although the OPLS-DA score scatter plot (Figure 3B) potentiated group separation, considerable group overlap was still visible, suggesting a certain degree of similarity in the metabolome signature before and after supplementation with cholecalciferol.

Model validation revealed the low accuracy of the model’s R^2^ parameter (<0.6) and the low predictive ability of the model (Q^2^ < 0.3) (Appendix A). However, despite its modest explanatory and predictive capacity, it was not overfitted (Appendix A), and we considered it for the identification of metabolites responsible for metabolome changes induced by cholecalciferol supplementation. OPLS-DA VIP values (VIP > 1) combined with the paired univariate analysis provided insight into metabolic modulation by vitamin D (Figure 4). As can be seen, various classes of metabolites were affected by vitamin D therapy, including amino acids, as well as glucose and lipid metabolism.

One of the hallmarks of hypertension identified via the analysis of this specific patient cohort was a systemic decrease in the levels of several blood amino acids. After 24 weeks of exposure to cholecalciferol, this trend was partially reverted for some of them. The most notable changes were observed for glutamine levels, which after the treatment increased towards the levels observed in normotensive patients (Figure 4), although they did not reach their baseline levels (Appendix A). Histidine levels also increased, but the relative weights of other glucogenic amino acids such as serine and threonine, which were high in week 0, decreased in the follow-up, as demonstrated by the univariate analysis (Appendix A). Ketogenic amino acids remained stable during the trial, with changes observed only in the univariate analysis of lysine levels (Appendix A).

At the same time, the levels of glucose decreased in patients treated with vitamin D (Figure 4). A decrease was also observed in the acetate levels. Acetate is at the crossroads of the glucose and the lipid metabolism and links those pathways to oxidative phosphorylation via the Krebs cycle.

Incomplete suppression of macromolecule signals by the 1D CPMG NMR sequence permitted the analysis of blood lipids. Although finer differentiation between the lipid species present in blood was not possible under the experimental conditions, we were able to observe changes in aliphatic chain methylene (CH_2_, 1.25 ppm) and olefinic hydrogen (CH = CH, 5.3 ppm), suggesting changes in the length and saturation of fatty acids. A lack of change in the signals corresponding to terminal CH_3_ groups of fatty acids (0.85 ppm) indicated that the total lipid levels remained unchanged. While it was not possible to fully characterize the affected lipid species, the statistically significant increase that was observed (Figure 4) may suggest a role of liposoluble vitamin D in lipid metabolism. Further changes included perturbations in creatine metabolism, specifically a statistically significant increase in creatinine levels after vitamin D, suggesting increased muscle mass in obese hypertensives.

When compared with the normotensive controls, perturbed amino acid levels remained the main distinction between hypertensive and normotensive obese/overweight subjects, even after the long-term vitamin D treatment of hypertensives (Figure 5).

In parallel, in the hypertensive control group, (HyperCG24) no significant changes in circulating metabolite levels were documented at 24 weeks (Appendix A). The combination of complete overlap in the PCA and the overfitted OPLS-DA model suggested that the blood metabolome did not suffer any changes during this study (Appendix A). Given that physiological variations are constantly occurring, this control group allowed us to track changes that were consequences of normal physiological processes rather than the vitamin D treatment. In our trial, no modifications were seen in the follow-up for the control hypertensive patients (HyperCG), further supporting the role of vitamin D as a modulator of the metabolic pathways.

## 3. Discussion

In this trial, we found that cholecalciferol supplementation and greater vitamin D bioavailability can induce changes in the serum metabolome, especially in glucose and glutamine levels. Although a generalized catabolic state persisted and no changes in the insulin-resistance indexes were observed, the microenvironment that surrounds peripheral cells, such as immune cells, was modulated, which may have had important biological effects.

Despite successful BP control in obese hypertensives, differences observed in the basal systemic metabolome when compared with normotensives reflect the complex interplay of hypertension and obesity with medicinal interventions and various other confounding factors. Our data confirm increased skeletal muscle catabolism, even in treated hypertension, as previously shown by others [24,25], with a significant downregulation of amino acid levels that is still observed after cholecalciferol. Pharmacometabolomic studies of hypertension are still rare, and the available data mostly describe the influence of pharmacologic interventions on lipid metabolism in the non-overweight population with early hypertension onset [26] or the discovery of metabolic markers in relation to specific medications [27].

On the other hand, decreased acetate and lactate levels after treatment suggest that improved metabolic health may ensue after cholecalciferol since elevated lactate is a marker of impaired aerobic metabolism and an indicator of the severity of metabolic syndrome [28]. Therefore, our results may point to the benefits of the combination of anti-hypertensives and vitamin D. Apart from the control of hypertension carried out by blood pressure-lowering drugs, the addition of cholecalciferol might have an overall beneficial effect on glucose metabolism. Given this ability to reduce glucose availability, combination with some anti-hypertensives, such as diuretics or β-blockers, may represent a plausible strategy to improve metabolic health [29]. Our trial failed to document differences in insulin-resistance indexes, which are indirect measures of long-term glycemic control, but other clinical studies have established the positive effects of cholecalciferol in type 2 diabetes [30], pointing to reduced gluconeogenesis as the mechanism responsible for the decrease in glucose [31,32]. In addition, the use of more sensitive measures, such as the homeostatic model assessment index, would probably better reflect the observed changes at the metabolome level. It is also relevant to point to the role of the mediators of the renin–angiotensin system in insulin-resistance modulation [33], even if in our trial no baseline or follow-up differences were documented in the prescription of these drugs in both groups.

The baseline levels of vitamin D and HbA1c should also be taken into account when analyzing the effects of the intervention on the outcomes. Very low levels of vitamin D were documented, and, considering the patient population treated, higher levels would be difficult to attain using standard doses of cholecalciferol. On the other hand, pre-trial levels of HbA1c were also not very high but were associated with the high visceral adiposity index. In this setting, the finding of lower systemic glucose levels points to the beneficial role of vitamin D even in the presence of high insulin resistance and long-lasting comorbidities.

In another trial of healthy patients [34], the serum metabolome 4 weeks post-cholecalciferol demonstrated changes not only in glucose but also in glutamine. We also found that vitamin D induced an increase in glutamine towards the levels observed in normotensive subjects. Previously, a glutamine increase after vitamin D supplementation was observed in vitamin D-responsive individuals, unraveling a distinct and healthier metabotype in terms of insulin-resistance and inflammatory markers [34]. These increased levels also represent greater availability to peripheral cells, especially circulating immune cells, which are avid consumers of this micronutrient for adenosine triphosphate (ATP) production [35,36]. Since hypertension is recognized more and more as an inflammatory disease, where immune cells have a pivotal role in the triggering and perpetuation of chronic and low-grade inflammation, cholecalciferol may foster lower activation and proliferation of immune cells.

Glutamine is fundamental for T-cell function, and its uptake increases upon T-cell activation [37], leading to full differentiation and activation. Increased levels of glutamine are essential to support highly proliferative cells, such as immune cells, probably contributing to a more efficacious immune cell response that is blunted when glutamine transporters or metabolism enzymes are blocked or when glutamine levels are low [37]. However, the need for nutrients and the subsequent outcomes are also cell-dependent. The response of cells to glutamine is a complex and dynamic process that includes signaling pathways that regulate glutamine synthesis and the rate of glutaminolysis, and its utilization can also depend on the availability of other energy substrates, such as glucose [38].

In this way, the metabolic remodeling induced by vitamin D, and the consequent differential expression of several micronutrients, may lead to a re-routing of several pathways in distinct cell populations. In T-cell subsets, for example, which are highly dependent on glucose and glutamine, these changes may induce reshaping and changes in the pattern of activation and cytokine secretion. In the case of short-lived Th17 cells that mostly rely on aerobic glycolysis and glutaminolysis for ATP production [39], increased glutamine may be responsible for Th17 cell stability and the modulation of reactive oxygen species synthesis and oxidative stress [39]. Although our trial was not designed to evaluate specific metabolic pathways or, specifically, the uptake of glutamine at the cellular level, we previously documented a restrictive effect of cholecalciferol on Th17 cell expansion in obese hypertensives [22]. In type 2 diabetics, for example, a profound inhibition of aerobic glycolysis in human T cells under the effect of vitamin D was also observed, contributing to a shift in the metabolic fate of immune cells from an activated state to an oligo-secreting state [40]. Likewise, these effects were documented in cells of innate immunity [41] and probably contribute to chronic inflammation modulation.

Vitamin D also affected the levels of other amino acids, including some branched-chain (BCAAs) and aromatic amino acids as well as lysine and threonine, and their role in the context of insulin resistance and the associated complications is well described [42]. For example, BCAAs are usually found to be increased in diabetes and obesity [43], probably due to defective oxidative metabolism [44], and to be correlated with the severity of insulin resistance [45]. Although BCAAs are fundamental for metabolic health, increased levels are consistently described as mediators of insulin resistance [44], and decreased expression of the enzymes of BCAA metabolism is described in adipose tissue [46]. In two recent trials, vitamin D reduced the levels of BCAAs [47] and promoted the catabolism of BCAAs in immune cells [48].

The analysis of the NMR spectra also points to changes in lipid composition in this cohort. The short-term influence of vitamin D on various classes of phospholipids and acylcarnitine was previously described in critically ill patients and was linked to decreased mortality [18] and fatty acid oxidation in adipose tissue [49]. Considering the limitations of the analytical technique used in this study, the relevance of these findings should be validated using more appropriate methods.

In addition, several other limitations should be mentioned. As previously described, the trial was designed to explore the differences in the frequency of T regulatory cells (Tregs). The sample size calculation was based on the average percentage of Tregs in a healthy population, and a sample of 36 participants fit these demands. Of course, a greater sample size would have provided more information, especially for the distinction between physiological and therapeutically driven changes, and metabolomic studies usually encompass significantly more participants. However, in the setting of a single-center trial, this also presented a recruitment challenge. To overcome this limitation, data from two control groups were added, allowing for better comprehension of the changes observed in the cholecalciferol-treated patients. The 24-week levels of vitamin D are also an important caveat, but the determination of free vitamin D helped overcome these limitations, as sufficient levels were documented. However, even in a setting with high doses, some benefits may be dependent on the patient’s underlying clinical condition [50], and vitamin D correction is a hard task in the setting of obesity. Possible confounding effects of anti-hypertensives, anti-diabetic medications, and statins should also be appreciated, considering the differential prescription of statins and the risk of hyperglycemia among the different classes of BP-lowering drugs. Finally, only Caucasian patients were included, which limits the translation of the obtained results to other patient populations as well as non-obese hypertensives.

In conclusion, the complex interrelationship of the different metabolites in obesity and hypertension has been studied in several studies with variable scales, but we are far from definite conclusions [51]. At least for cholecalciferol, a role in reducing the availability of glucose and enhancing glutamine levels suggests that, in the long term, it may be a useful adjunctive therapy to modulate chronic inflammatory conditions.

## 4. Materials and Methods

### 4.1. Trial Design and Patient Population

The results presented here are part of a prospective, single-center, randomized clinical trial conducted in patients with obesity-associated hypertension. The detailed study design has been published elsewhere [22]. Briefly, 36 patients routinely followed in the hypertension consultation with simultaneous obesity or metabolic syndrome diagnoses were recruited to a phase 2, open, parallel-group trial designed to explore the effects of cholecalciferol on the serum metabolome. Since several metabolic pathways are disturbed during obesity-related hypertension, most frequently glucose, lipid, and amino acid pathways, we chose to evaluate the differential expression of the metabolites related to those pathways. This included the study of different subclasses of amino acids, namely glucogenic, ketogenic, and both gluco- and ketogenic amino acids, with a special focus on glutamine, given its anabolic properties and importance as a source of substrates for the synthesis of other metabolites. We also included an evaluation of glucose and subproducts of glucose metabolism, searching for metabolites such as pyruvate, lactate, and acetate as well as other Krebs cycle intermediates that could point to the preferential routes of glucose utilization. Finally, lipids were also studied, and although the technique we employed may have had some limitations in the identification of different lipid-derived metabolites, it allowed the distinction of free fatty acids, their concentrations, and an evaluation of the differential degree of saturation.

The endpoint was thus defined as the changes in the serum concentrations of metabolites derived from each of the selected pathways: amino acids, glucose, and lipids.

For endpoint evaluation, patients were selected according to age, gender, and body mass index (BMI) criteria. Patients had to have an established diagnosis of essential hypertension and no suspected contribution of secondary causes other than the simultaneous diagnosis of obesity (BMI ≥ 30 Kg/m^2^). Patients that did not meet these two criteria but had superimposed metabolic syndrome, as defined by the International Diabetes Federation [52], were also allowed to be included because of their comparable pathophysiology and the well-recognized continuum between metabolic syndrome and obesity. Equal numbers of men and women were recruited to avoid gender-specific metabotype expression bias. Concerning the burden of blood pressure-lowering drugs, patients could have up to four anti-hypertensives prescribed, independent of the anti-hypertensive class.

The exclusion criteria also included vitamin D > 20 ng/mL, immune-inflammatory conditions other than obesity, hypercalcemia-prone diseases, and baseline hypercalcemia. Ongoing infectious and neoplastic diseases, as well as glomerular filtration rates < 60 mL/min/1.73 m^2^ according to the Modification of Diet in Renal Disease Study Equation, also precluded patients’ inclusion.

The trial included a 12-week screening period, after which stratified randomization and three planned visits ensued. Patients were divided into four blocks (men versus women and diabetics versus non-diabetics) and randomized in a 1:1 ratio, avoiding significant mismatches that could contribute to metabolite differences. Trial allocation was concealed using a random sequence for each block, and the patients were assigned to the control group or the cholecalciferol intervention via sealed envelopes. All samples were coded, and the laboratory technicians were blinded to their group assignments.

The baseline visit occurred during a summer month for all participants, specifically July, to maximize sun exposure. This was followed by two planned visits: a visit in week 16 for cholecalciferol dose adjustment in the vitamin D group and a visit in week 24 for endpoint assessment.

The experimental design complied with the ethical principles laid out in the 2008 revision of the Declaration of Helsinki and was approved by an ethics committee prior to launching the study. Approval from national and local competent authorities (National Agency for Medicines and Health Products (INFARMED), National Ethics Committee for Clinical Investigation (CEIC), and Local Health Unit of Castelo Branco Ethics Committee) was obtained (Portugal-based National Clinical Trials Registry No. MD001501). Informed consent was obtained from all subjects recruited to the trial. An independent data safety monitoring board regularly reviewed all trial procedures and the safety of the patients.

### 4.2. Cholecalciferol Prescription and Vitamin D Monitoring

Cholecalciferol was provided in bottles of 25,000 IU each. The recruited patients had vitamin D concentrations up to 20 ng/mL, for which cholecalciferol was prescribed as follows: 50,000 IU/week for 8 weeks. This dose was followed by 25,000 IU every other week until week 16. At this time point, the serum vitamin D measurement was repeated, and the cholecalciferol prescription was adjusted. The posology scheme followed the recommendations for the general population after a high-dose supplementation period: if the vitamin D concentration was still ≤20 ng/mL at 16 weeks, the cholecalciferol dose was 50,000 IU/week until the end of the trial, and if the vitamin D concentration was between 20 and 30 ng/mL, one bottle every 4 weeks was prescribed. For those patients with higher levels at 16 weeks, the cholecalciferol dose was reduced to 25,000 IU every 6 weeks. The patients were instructed to take the experimental drug at a principal meal and to store it away from light.

### 4.3. Endpoint Assessment

The blood sampling for the metabolomic analysis was performed in weeks 0 (before vitamin D supplementation) and 24 (at the end of the trial). An additional control group that included seventeen normotensives age- and gender-matched with the trial participants was also evaluated in week 0. These controls were overweight but non-diabetic, and ten of them also had dyslipidemia. As this was a single-center study, all participants and controls were recruited from the same hospital.

The trial also included blood sampling for a hemogram, total and free vitamin D, a lipid profile, measurement of insulin-resistance indexes (HbA1c and visceral adiposity index (Appendix A)), and metabolomic analysis. On the day of collection, samples obtained for metabolome evaluation were immediately frozen at −20 °C at the trial center and on the same day were transported to the central laboratory (CICS-UBI), where storage proceeded at −80 °C until the analysis was performed. Per the protocol, the endpoint was only evaluated at the end of the trial in both groups, and no additional sampling was performed.

### 4.4. Nuclear Magnetic Resonance-Based Metabolomics

A metabolomics approach using nuclear magnetic resonance (NMR) was used to evaluate changes in the total metabolome of blood serum. Plasma samples obtained from both obese/overweight normotensive controls and obese/overweight hypertensive trial participants were analyzed at 0 and 24 weeks. The processing of the samples included thawing, homogenizing using a vortex, and centrifuging (9200 rpm for 5 min). Only small metabolites were analyzed since large macromolecules, such as proteins and lipids, were filtered out by applying a 1D Car–Purcell–Meiboom–Gill (CPMG) NMR experiment. A 14.1 T Varian Inova 600 MHz (Agilent Technologies, Santa Clara, CA, USA) spectrometer was used for NMR spectra acquisition. This spectrometer was equipped with a 3 mm QXI probe and z-water presaturation. This allowed for measurement of water saturation frequencies and adjustment for each sample. The shimming reference used was sodium fumarate at a final concentration of 1 mM. ^1^H cpmg (Carr–Purcell–Meiboom–Gill spin-echo pulse sequence) spectrum was acquired for each sample using the following conditions: 298 K, 9.6 kHz spectral width, 90° pulse angle, 4 s relaxation delay, and 3.25 s acquisition time. In total, 128 scans were collected, with 8 dummy scans, total spin–spin relaxation delay of 0.75 ms and 374 loops. Spectra were processed by applying exponential line broadening (lb 0.3 Hz). For each one, the processed spectrum phase and base line were manually adjusted.

Using a glucose doublet at 5.23 ppm, the chemical shifts were internally referenced. The public database HMBD was used for metabolite assignment and was accessed on 6 December 2022. Based on the available reference spectra, 2D homonuclear TOCSY spectra were recorded to help with spectral assignment. For multivariate analysis, data matrices were created in AMIX-Viewer (version 3.9.15, BrukerBiospin, Rheinstetten, Germany) by integrating metabolite multiplets. Integrals were normalized based on the total integral area to account for the experimental conditions [53].

### 4.5. Statistical Analysis

Obtained normalized areas representative of metabolite concentrations in human serum were analyzed by univariate and multivariate statistical analyses. A multivariate approach based on data reduction was used to analyse metabolite abundance (30 dependent variables in total). NMR spectra of 87 serum samples were included in the analysis based on the spectral quality. Principal component analysis (PCA) was used to visualize the initial data structure, evaluate possible grouping, and identify outliers. Next, a supervised method, orthogonal projection to latent structures discriminant analysis (OPLS-DA) was applied to pair-wise group comparisons to identify differentially expressed metabolites with discriminative potential. OPLS-DA models were cross-validated by permutation test using 100 permutations, and the permuted parameters R2Y and Q2 were used to assess the fitting validity and predictive abilities of the models (*p* < 0.05 for comparison of each permuted parameter with the original classification was deemed statistically relevant). Models with similar values of R2 and Q2 (within 0.2–0.3 units), Q2 > 0.5, and validated by permutation test were considered relevant. Metabolites with OPLS-DA VIP > 1 were considered relevant to group separation. In PCA and OPLS-DA plots, ellipsoid curves represent 95% confidence intervals. Autoscaling of variables (metabolites) was applied for multivariate analysis. For univariate analysis, a non-parametric, independent samples test, with false discovery rate (*p*-value threshold 0.1) was used to evaluate differences between normotensive controls (Normo0) and hypertensive patients (HyperAG0 and HyperCG0) before the vitamin D treatment as well as to evaluate metabolome differences between HyperAG0 and HyperCG0. For comparisons of active treatment (HyperAG) and control hypertension (HyperCG) groups before and after 24 weeks of treatment, a paired comparison non-parametric Wilcoxon rank-sum test with FDR (*p* < 0.1) was applied. Univariate and multivariate analyses were performed on online platform MetaboAnalyst 5.0 accessed on 6 December 2023 (www.metaboanalyst.ca) [54]. Box-whisker plots based on medians with interquartile range were produced by GraphPad Prism 8.0.1 (GraphPad Software Inc., Boston, MA, USA).

## 5. Conclusions

The complex interrelationship of the different metabolites in obesity and hypertension has been studied in several studies with variable scales, but we are far from definite conclusions. At least for cholecalciferol, a role in reducing the availability of glucose and enhancing glutamine levels suggests that, in the long term, it may be a useful adjunctive therapy to modulate chronic inflammatory conditions. Given its safety profile when compared to available immunomodulators, low cost, and widespread availability, the potential of cholecalciferol to modify serum metabolome arises as a novel and trustworthy tool to add to the obesity-related hypertension options.

## Figures and Tables

**Figure 1 ijms-25-03416-f001:**
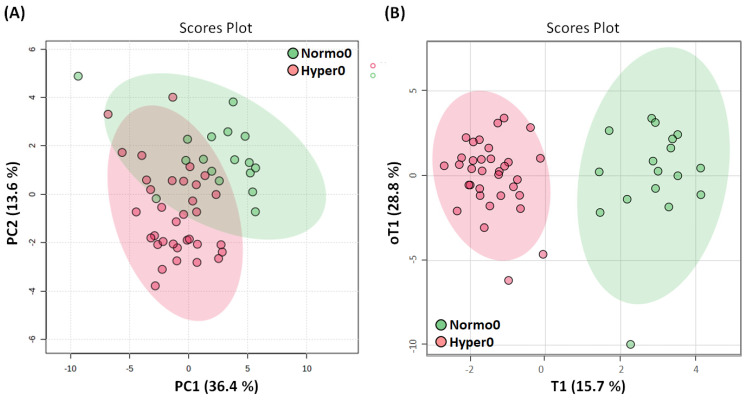
Blood metabolome of normotensive controls (Normo0, n = 17) and hypertension patients (Hyper0, n = 34) before vitamin D therapy. (**A**) PCA scores scatter plot of blood serum 1D Car–Purcell–Meiboom-Gill (CPMG) nuclear magnetic resonance (NMR) spectra. PC1 and PC2 and related explained variabilities (%) are indicated on axes x and y, respectively. (**B**) OPLS-DA scores scatter plot of blood serum 1D CPMG NMR spectra. The first predictive (T1) and the first orthogonal (oT1) components and their respective explained between- and within-group variabilities (%) are indicated on axes x and y. The model is described by 1 predictive and 3 orthogonal components, with total explained variation values of R^2^X = 0.58 and R^2^Y = 0.89 and a goodness of prediction (Q^2^) of 0.73. Each point in the scatter plots represents a single patient as a function of all analyzed metabolites, and 95% confidence intervals are represented by ellipses.

**Figure 2 ijms-25-03416-f002:**
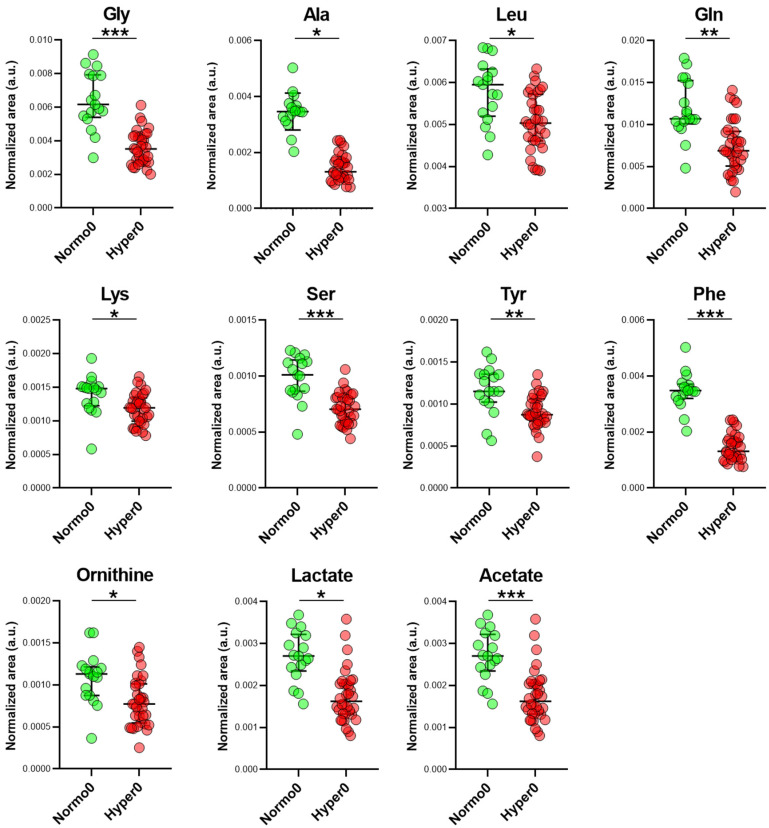
Differentially expressed metabolites between normotensive (Normo0, n = 17) and hypertensive patients (Hyper0, n = 34) before vitamin D therapy (t0). The differentially expressed metabolites between the 2 groups were selected based on the OPLS-DA VIP values (>1) and univariate analysis (*p* < 0.05). Medians with interquartile ranges are presented. Statistical significance values obtained via univariate non-parametric tests are denoted as follows: * *p* < 0.05, ** *p* < 0.001, *** *p* < 0.0001.

**Figure 3 ijms-25-03416-f003:**
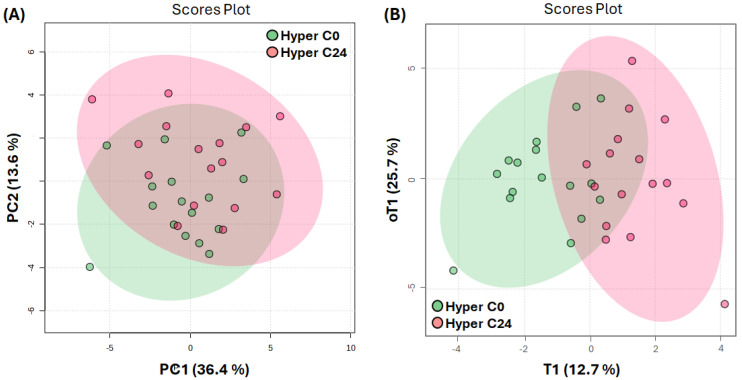
Comparison of blood metabolome in hypertensive active therapy group before the therapy (Hyper-AG0, n = 16) and after 24 weeks of vitamin D treatment (Hyper-AG24, n = 16). (**A**) PCA scores scatter plot of blood serum 1D Car–Purcell–Meiboom–Gill (CPMG) nuclear magnetic resonance (NMR) spectra. PC1 and PC2 and related explained variabilities (%) are indicated on axes x and y, respectively. (**B**) OPLS-DA scores scatter plot of blood serum 1D CPMG NMR spectra. The first predictive (T1) and the first orthogonal (oT1) components and their respective explained variabilities (%) are indicated on axes x and y. The model is described by 1 predictive and 1 orthogonal component, with total explained variation values of R^2^X = 0.38 and R^2^Y = 0.59 and a goodness of prediction (Q^2^) of 0.28. Each point in the scatter plots represents a single patient as a function of all analyzed metabolites, and 95% confidence intervals are represented by ellipses.

**Figure 4 ijms-25-03416-f004:**
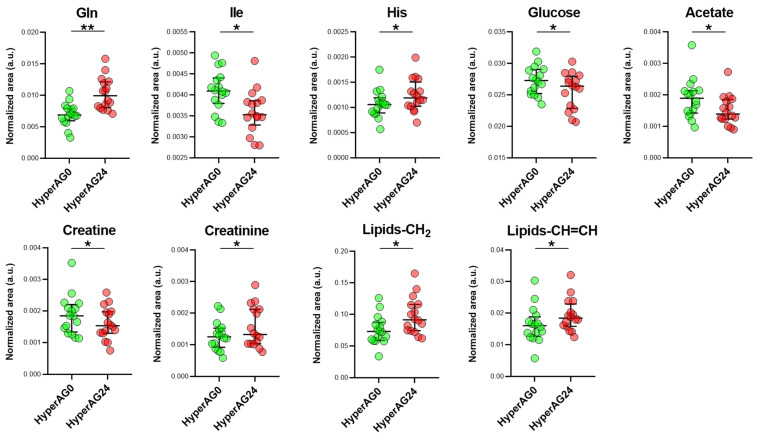
Metabolome changes induced by vitamin D therapy. Differentially expressed metabolites in hypertensive patients before (Hyper-AG0, n = 16) and after vitamin D therapy (Hyper-AG24, n = 16). The differentially expressed metabolites between the 2 groups were selected based on the OPLS-DA VIP values (>1) and the paired univariate analysis (*p* < 0.05). Medians with interquartile ranges are presented. Statistical significance values obtained via univariate non-parametric tests are denoted as follows: * *p* < 0.05, ** *p* < 0.001.

**Figure 5 ijms-25-03416-f005:**
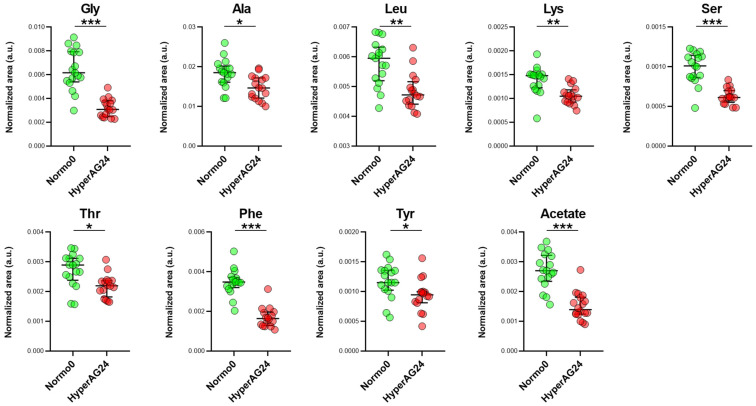
Differentially expressed metabolites between normotensive patients (Normo0, n = 17) and hypertensive patients after vitamin D therapy (HyperAG24, n = 16). The differentially expressed metabolites between the two groups were selected based on the OPLS-DA VIP values (>1) and the univariate analysis (*p* < 0.05). Medians with interquartile ranges are presented. Statistical significance values obtained via univariate non-parametric tests are denoted as follows: * *p* < 0.05, ** *p* < 0.001, *** *p* < 0.0001.

**Table 1 ijms-25-03416-t001:** Baseline characteristics of trial participants (average ± SD).

	Week 0		
	Non-Cholecalciferol Group	Cholecalciferol Group	*p*-Value
Gender (men/women)	9/9	9/9	--
Vitamin D (ng/mL)	17.2 ± 5.6	14.4 ± 4.3	0.100
Free vitamin D (pg/mL)	6 ± 2.4	5.2 ± 1.2	0.210
Visceral adiposity index	5.7 ± 4.1	5.6 ± 3.4	0.957
HbA1c (%)	5.8 ± 0.7	6.1 ± 1.1	0.416
BMI (Kg/m^2^)	35.1 ± 5.9	34 ± 4.3	0.502
Waist circumference (cm)	117.7 ± 8.7	110 ± 11.3 *	0.026
Systolic BP (mmHg)	138.8 ± 19.8	131.3 ± 15	0.210
Diastolic BP (mmHg)	84.2 ± 9.1	79 ± 10.9	0.133
Number of diabetic patients	5	6	0.727
Total cholesterol (mg/dL)	164.2 ± 34.8	164.6 ± 27.5	0.966
HDL cholesterol (mg/dL)	47.4 ± 12.5	48.2 ± 12.3	0.842
Triglycerides (mg/dL)	137.7 ± 73.7	147.1 ± 63.1	0.684
Patients on statin therapy	11	17 *	0.015
Number of AH drugs	2.8 ± 1	2.3 ± 0.9	0.146
ACEI/ARA	16	16	0.992
CCB	4	8	0.747
Beta-blockers	9	5	0.087
Diuretic	16	14	0.218
α-blockers	1	0	0.324
Estimated GFR	100.4 ± 13.1	103.3 ± 11.7	0.572

HbA1c: glycated hemoglobin; BMI: body mass index; BP: blood pressure; HDL: high-density lipoprotein; AH: anti-hypertensive; ACEI: angiotensin-converting enzyme inhibitor; ARA: angiotensin receptor antagonist; CCB: calcium channel blocker; GFR: glomerular filtration rate. Statistical significance of univariate parametric tests is denoted as follows: * *p* < 0.05.

**Table 2 ijms-25-03416-t002:** Baseline characteristics of normotensive controls and trial participants (average ± SD).

	Week 0		
	Normotensive Controls (N = 17)	Trial Participants (N = 36)	*p*-Value
BMI (Kg/m^2^)	27.8 ± 4.1	34.5 ± 5.1 **	0.0001
Systolic BP (mmHg)	117.8 ± 5.6	135.6 ± 18.2 **	0.0003
Diastolic BP (mmHg)	75.8 ± 5.1	81.9 ± 10.2 *	0.024
HbA1c (%)	--	5.9 ± 0.89	
Total cholesterol (mg(dL)	183.5 ± 36.4	164.4 ± 30.9	0.053
HDL cholesterol (mg/dL)	47.4 ± 12.5	48.2 ± 12.3	0.842
Triglycerides (mg/dL)	113.8 ± 41.2	142.4 ± 67.8	0.115
Patients on statins (%)	58.8%	77.7%	0.621
Estimated GFR (mL/min/1.73 m^2^)	115.2 ± 34.5	101.8 ± 35.6	0.202

BMI: body mass index; BP: blood pressure; HbA1c: glycated hemoglobin; HDL: high-density lipoprotein; GFR: glomerular filtration rate. Statistical significance of univariate parametric tests is denoted as follows: * *p* < 0.05, ** *p* < 0.001.

**Table 3 ijms-25-03416-t003:** Patients’ characteristics at 24 weeks (average ± SD).

	Week 24		
	Non-Cholecalciferol Group	Cholecalciferol Group	*p*-Value
Vitamin D (ng/mL)	15.5 ± 3.4	26.9 ± 5.6 **	<0.001
Free vitamin D (pg/mL)	4.6 ± 2.3	6.2 ± 2.3 *	0.044
Visceral adiposity index	5.4 ± 4.1	5.2 ± 3.8	0.869
HbA1c (%)	5.6 ± 0.5	6.1 ± 1.1	0.347
BMI (Kg/m^2^)	34.9 ± 6.1	34.5 ± 4.7	0.763
Waist circumference (cm)	114 ± 11	107 ± 9.9	0.049
Systolic BP (mmHg)	137 ± 17	133.4 ± 15.3	0.356
Diastolic BP (mmHg)	84.5 ± 8.8	81 ± 9.8	0.176

HbA1c: glycated hemoglobin; BMI: body mass index; BP: blood pressure. Statistical significance of univariate parametric tests is denoted as follows: * *p* < 0.05, ** *p* < 0.001.

## Data Availability

The data presented in this study are available on request from the corresponding author. The data are not publicly available due to ethical reasons.

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
