# Peer review of "Standard Doses of Cholecalciferol Reduce Glucose and Increase Glutamine in Obesity-Related Hypertension: Results of a Randomized Trial"

_ijms, 2024, doi:10.3390/ijms25063416_

Round 1
Reviewer 1 Report
Comments and Suggestions for Authors
As the subjects in the control group are shown to be vitamin D deficient, how could it be justified that they were not treated with proper supplementation through this rather long period?
What did the authors mean by "usual therapy" for the control group?
How did the researchers classify the overweight normotensive subjects as healthy controls? Were they "metabolically healthy"?
Please pay proper attention to the use of appropriate statistical phrases. Multivariate approaches need more than one dependent variable. Please clarify.
In Table 1 the levels of HDL-C seem to be incorrect.
The baseline levels of vitamin D and glucose should be taken into account when analyzing the effects of the intervention on the outcomes.
Comments on the Quality of English Language
The English language of the manuscript needs to be improved.
Reviewer 2 Report
Comments and Suggestions for Authors
The manuscript describes the evaluation of the serum metabolome of 36 age- and gender-matched adults, with obesity related hypertension and vitamin D deficiency, before and after supplementation with cholecalciferol along the usual therapy, towards exploring the influence of cholecalciferol on metabolic pathways as a possible mechanism of its immunomodulatory activity in obesity-related hypertension.
The topic is of practical significance, the scope is limited. The title should be edited.
Reproducibility is compromised in that some instrument’s full product information is missing, such as the NMR instrument; statistical analysis should be a separate section and the applied alpha value, confidence interval should be introduced.
Further, as this study is “a secondary goal of a phase-2, randomised, single centre based, 24-week trial”, this may undermine the significance and validity of the study, in that the study design, including but not limited to hypotheses, independent and dependent variables, experimental design, data collection methods and a statistical analysis plan, etc., beyond the sample size and inclusion criteria already discussed, may have not been optimal for achieving this goal. Also the authors need to justify how it does not fall within the practice of incremental “salami science” which is roughly defined as a publication of two or more articles derived from a single study.
This, compounded with the inconclusive conclusion section, makes one wonder if this work is worthy of publication, before a properly designed and executed experiment is carried out towards drawing a definite conclusion. Although the honesty is appreciated, not all work deserves to be published in IJMS.
Minor English editing is needed.
Hence, major revision is recommended before further conclusion, also addressing the following additional points:
Result presentation needs to be modified: In Fig. 2,4,5, * should be put in between groups rather than on top of one group. Fig. 5 what does # indicate?
Conclusion should not include references.
Line 317-321 recommend also cite work An ab initio exploratory study of side chain conformations for selected backbone conformations of N-acetyl-l-glutamine-N-methylamide 2001.
Comments on the Quality of English Languageminor editing
Round 2
Reviewer 2 Report
Comments and Suggestions for Authors
After the revision and clarification, all concerns have been addressed.